CUTTING EDGE

# Lessons from Fraxinus, a crowd-sourced citizen science game in genomics

**Abstract** In 2013, in response to an epidemic of ash dieback disease in England the previous year, we launched a Facebook-based game called Fraxinus to enable non-scientists to contribute to genomics studies of the pathogen that causes the disease and the ash trees that are devastated by it. Over a period of 51 weeks players were able to match computational alignments of genetic sequences in 78% of cases, and to improve them in 15% of cases. We also found that most players were only transiently interested in the game, and that the majority of the work done was performed by a small group of dedicated players. Based on our experiences we have built a linear model for the length of time that contributors are likely to donate to a crowd-sourced citizen science project. This model could serve a guide for the design and implementation of future crowd-sourced citizen science initiatives.

GHANASYAM RALLAPALLI, FRAXINUS PLAYERS, DIANE GO SAUNDERS, KENTARO YOSHIDA, ANNE EDWARDS, CARLOS A LUGO, STEVE COLLIN, BERNARDO CLAVIJO, MANUEL CORPAS, DAVID SWARBRECK, MATTHEW CLARK, J ALLAN DOWNIE, SOPHIEN KAMOUN, TEAM COOPER AND DAN MACLEAN

## Introduction

Ash dieback is a disease caused by the fungal pathogen *Hymenoscyphus fraxineus*, and it has devastated populations of ash trees (*Fraxinus excelsior*) across Europe in recent years. When ash dieback was discovered in the wild in the east of England for the first time, in 2012, the present authors set up the OpenAshDieBack (OADB) project as a crowdsourcing platform to allow scientists across the world to contribute to the genomic analysis of the pathogen and the host (*MacLean et al., 2013*). Subsequently, we developed and released Fraxinus, a Facebook-based game, to allow non-specialists to improve genetic variant predictions from DNA sequence data arising from the OADB project (*MacLean, 2013*).

DNA sequence alignment is a hard problem that seeks to arrange two or more genome sequences in order to identify regions of similarity. When short fragments are being aligned with a longer sequence, the longer sequence is often considered to be a reference sequence that should not be altered. The process of alignment requires the best overall match between the two sequences to be found first: this 'global alignment' is then followed by a finer-grained 'local alignment' that involves modifying the short sequences by, for example, inserting small gaps or deleting short stretches of the sequence.

Alignment is a computationally intensive process, and many computer programs (e.g., BWA aligner [*Li and Durbin, 2009*]) have been devised that implement and optimize alignments according to various measures of similarity. A straightforward measure is percent identity (in which the number of identical nucleotides in the alignment is calculated as a proportion of whole alignment length). Once the alignment process is complete, any differences between the two sequences can be considered a genetic variation. These differences can be single nucleotide polymorphisms (SNPs), in which a single nucleotide differs, or

*For correspondence:
dan.maclean@tsl.ac.uk

Present address: †Laboratory of Plant Genetics, Kobe University, Kobe, Japan

insertion–deletion polymorphisms (INDELs), in which longer stretches are different. Software like SAMTools (*Li et al., 2009*) can identify the variants from alignments.

Citizen-science projects are an excellent opportunity to engage the public in science and to harness the human intelligence of large numbers of non-specialists to make progress on inherently difficult research tasks. The citizen-science approach has been used in astronomy (*Lintott et al., 2008*), protein folding (*Cooper et al., 2010*), and genetics (*Kawrykow et al., 2012*; *Curtis, 2014*; *Ranard et al., 2014*). However, computational approaches to the identification of genetic variants can be error-prone, and it has been shown that the pattern-recognition skills of humans can improve DNA sequence alignments (*Kawrykow et al., 2012*). Therefore, we created Fraxinus to improve the automated alignments produced by computational approaches.

Over the first year, Fraxinus was loaded with variant calls based on cDNA-sequence reads generated from four different samples collected at locations in Norfolk in the UK: Kenninghall Wood (KW1), Ashwellthorpe Wood (AT1 and AT2), and Upton Broad and Marshes (UB1) (*Saunders et al., 2014*). The game presented the player with a pre-selected small section of genome sequence from the KW1 reference strain and rows of sequence from one of the other test variants (*MacLean, 2013*), with each DNA sequence represented as a string of colored leaves. The game provided tools that allow the player to shift the sequence relative to the reference and to edit the sequence in such a way as to introduce deletions or gaps. The aim of the game for the player was to produce the best alignment, allowing her/him to claim the puzzle as their own and score points. The game was played in Facebook and took advantage of the player's social network to invite new players. The

wider social network was used to encourage the replay of puzzles. We stored player names, scores, and the resulting alignments for later analysis. Here, we describe the Fraxinus game, the results of alignment comparisons, the response the game received, and details of parameters fitted to replicate player dynamics.

## Results and discussion

The BWA aligner (*Li and Durbin, 2009*) and SAMTools (*Li et al., 2009*) were used to identify SNPs and INDELs in AT1, AT2 and UB1 against the common reference KW1 (see 'Materials and methods'). Initially a data set of 1000 SNPs and 160 INDELs were loaded into Fraxinus (*Table 1*). For ease of playing, we limited the maximum number of sequences per puzzle to 20 (*Figure 1—figure supplement 1A*): in total 10,087 puzzles were created from the 1160 variants. Fraxinus was released on 13 August 2013 and by 4 August 2014 (51 weeks) had received 63,132 visits from 25,614 unique addresses in 135 countries (*Table 2*, *Figure 1—figure supplement 2*). Most of these were from the UK (57%), followed by the US, Canada, France, and Germany.

In the first 6 months, when the first batch of puzzles was retired, we received 154,038 alignment answers, with an approximate log-normal distribution of answers for 10,087 puzzles (*Figure 1—figure supplement 1B*). Each puzzle was played on average by 11.48 (standard deviation 7.08) players, and not all alignment answers were informative (we received 3.6 empty alignments per puzzle; *Supplementary file 1*, Table S1). We compared 7620 selected puzzles for differences in player and software alignments: in 4701 (61.7%) puzzles the alignments from all the high-scoring players differed from the computational alignment (*Figure 1A*). A further 2765 (36.3%) of the player alignment answers were

**Table 1**. Number of variants and derived puzzles used in Fraxinus version 1

| Fungal sample | SNP | | INDEL | |
| --- | --- | --- | --- | --- |
| | Variants | Puzzles | Variants | Puzzles |
| Ashwellthorpe1 | 250 | 2937 | 53 | 521 |
| Ashwellthorpe2 | 353 | 1121 | 51 | 170 |
| Upton broad and marshes1 | 397 | 4964 | 56 | 374 |
| Total | 1000 | 9022 | 160 | 1065 |

SNP, single nucleotide polymorphisms; INDEL, insertion–deletion polymorphism.

**Table 2**. Details about player visits and contributions made to Fraxinus

| Description | Details |
| --- | --- |
| Start date | 2013-08-12 |
| Date until | 2014-08-04 |
| Game duration in days | 358 |
| Total number of visits | 63,132 |
| Total number of players | 25,614 |
| Mean new visits % per day | 26.7 |
| Mean visit duration in minutes | 25.3 |
| Total time contributed in days | 924 |

identical to those from software. And with the exception of 154 puzzles (2%), the high-scoring players converged on the same answer (*Figure 1A*). Most puzzles had replicate analyses; 6834 (89.7%) had two or more high-scoring players (*Figure 1—figure supplement 1C*). Together, these results indicate a high accuracy of alignment across puzzles and endorse the notion that replay leads to useful replication.

In total, in 57,834 cases, the player alignment differed from the alignment produced by the computational method. In 15.26% of these cases, the players achieved a higher percent identity; in 6.37% of cases, the computational method achieved a higher percent identity; and the percent identities were equal for the remaining 78.37% of alignments (*Figure 1C*). Players aligned a higher proportion of INDELs (85.4% of 822) differently to software than SNPs (58.8% of 6798; *Figure 1B*). Players scored higher in 29% of INDEL alignments, with software scoring better in only 4% of alignments. For SNPs players scored better in 12% of puzzles with software scoring better in 7% (*Supplementary file 1*, Table S2).

The series of steps involved in these solutions is not yet recorded by Fraxinus, but previous work on the FoldIt game has shown that recording and sharing protein-folding recipes allow rapid development of novel algorithms (*Khatib et al., 2011*); we are hopeful that similar improvements will be possible with Fraxinus.

Most of the visitors to the Fraxinus game were interested only casually and did not play beyond either the introduction or tutorial (*Table 3*). Only 7357 (28.72%) of players answered puzzles and, more surprisingly, 49 players (0.7%) contributed to half of the answers (74,356

answers) that we received (*Figure 2A*). New players and returning players devoted an average of 12.5 min and 29.7 min per visit, respectively (*Figure 2—figure supplement 1A*). On average, each returning player made 2.3 visits per day, thereby contributing 70 min of game play per day; this indicates that the players were spending sufficient time to go through the process of realignment and are probably now experts in the alignment process.

As a high-profile project, a number of media-wide publicity events were organized to increase awareness of the game. There were clear peaks of new visitors (mean 151.4) on dates with publicity (*Figure 2B*) with events in traditional media having a much stronger impact than social media alone (*Supplementary file 1*, Table S3). The number of returning players showed only a small increase (mean 2.6 returning per publicity event; *Figure 2—figure supplement 1B*). Most (62%) players joining on a press release date played only on that day, and 97.5% of all players played for 10 days or less (*Figure 2—figure supplement 2A*). However, new players were more motivated, if they joined on a press release date and submitted 6.8 times more answers on average than those joining on other dates (*Figure 2—figure supplement 2B*). A marked increase in empty answers was also observed on press release dates (*Figure 2C*), indicating that new players joining on these days have contributed to more empty answers than useful answers. This suggests that care need to be taken in assuming that all time contributed by players is equally productive. In total, 88% of all answers were provided by players who joined within the first 30 days (*Figure 2D*). Therefore, most of the analysis in Fraxinus was carried out within the first few months by players enthused by the initial publicity, with some benefit resulting from subsequent publicity.

The number of daily visits followed a power law distribution, (*Figure 3—figure supplement 1*) in the period presented. The power law distribution is also followed by new players and returning players (*Figure 2—figure supplement 1C*). There were very few players who visited regularly (*Figure 3—figure supplement 2*). There are 33 players who were active on 60 or more days, while 8 of these were active for over 200 days (*Supplementary file 1*, Table S4), with one player being active for 332 out of 350 days, since joining. The distribution indicates that we

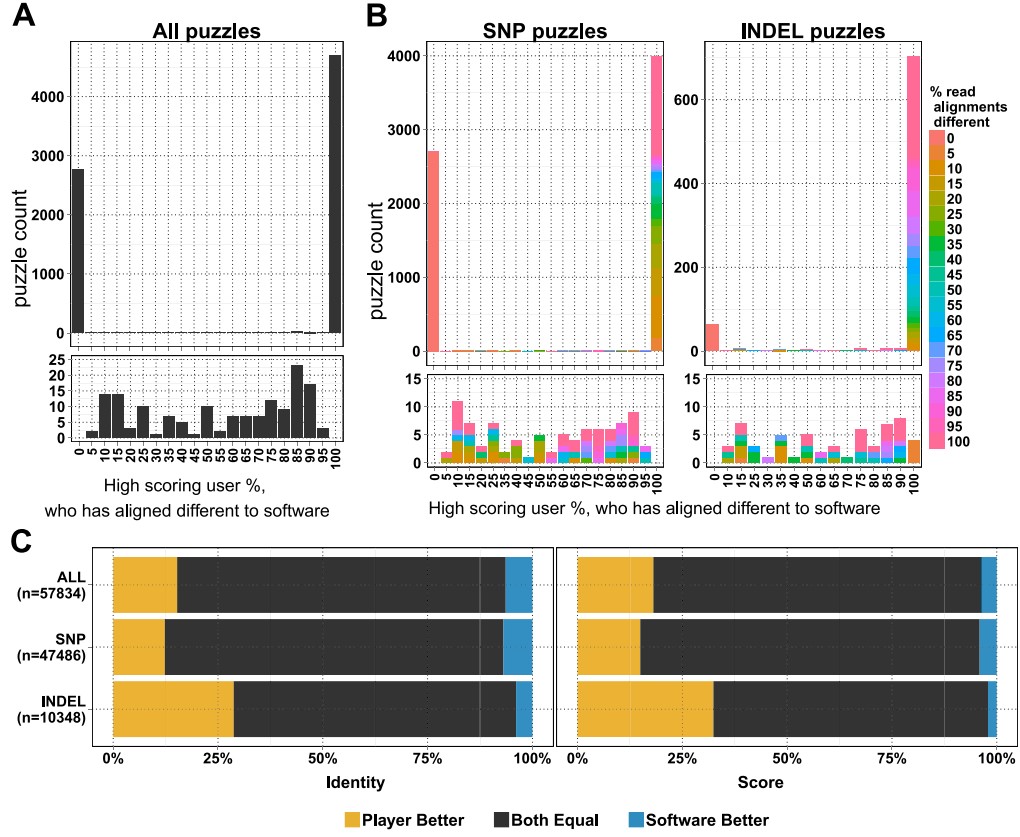

**Figure 1**. Comparison of player and software alignments for 7260 selected puzzles. (**A**) Number of puzzles (y-axis) vs percentage of high-scoring players who produce alignments *different* to the alignment produced by the BWA mem software (x-axis): in 4701 of these puzzles, the alignments produced by all of the high-scoring players were different to the alignment produced by the software (rightmost column; difference = 100%); in 2765 of puzzles, the alignments produced by all of the high-scoring players were the same as the alignment produced by the software (leftmost column; difference = 0%). Only a small number of puzzles (154) were between these two extremes (see lower panel, which expands the y-axis for differences between 5% and 95%). (**B**) Single nucleotide polymorphism (SNP) and insertion–deletion polymorphism (INDEL) puzzles presented separately and color coded with a heat map depicting the percent of read alignments contributing to the difference between player and software. (**C**) Comparison of alignments from the 4701 puzzles that had all high-scoring players aligned different to the software: the left panel is based on percent identity between sequences; the right panel is based on the Fraxinus game score (see *Fraxinus game setup* in 'Materials and methods').

The following figure supplements are available for figure 1:

**Figure supplement 1**. Number of reads, answers, and players per puzzle.

**Figure supplement 2**. Global distribution of Fraxinus game players; a number of visits are color coded.

**Figure supplement 3**. Selecting reads covering the variant allele.

can expect a surprisingly long life for Fraxinus in the order of years (*Figure 3A*). Despite appearing contradictory to the observation that most answers were submitted in the first few months, a small number of players are visiting frequently, although this is likely to decline in the future.

We examined whether success at the game affected longevity of contribution. The mean puzzles contributed per day for groups with different total game scores decreased in a linear fashion for all scoring bins, indicating that players' enthusiasm for the puzzles or cause decreased over time, irrespective of their

**Table 3**. Details about categories of players visiting Fraxinus

| Description | No. of players | Percent |
| --- | --- | --- |
| Viewed introduction | 6115 | 23.87 |
| Completed tutorial and scanned puzzles | 7958 | 31.07 |
| Attempted puzzles | 4184 | 16.33 |
| Scored puzzles | 7357 | 28.72 |
| Total players | 25,614 | 100 |

success at the game (*Figure 3B*). Using these demographic parameters as a base, we developed a simple model to predict the productivity of any proposed crowdsourcing project. The model returns the number of players contributing to a project per day, based on an initial expected cohort of players, a returning player rate, and a new player rate. We can then calculate the work ultimately done for a task that takes a given amount of time to learn and execute. Our model is formulated as a decreasing power law relationship ('Materials and methods') and allows for arbitrary increase in players as per those observed on press release days. The model accordingly recapitulates the observed result from Fraxinus (*Figure 3C*). Predicted work time in the model was 935.53 days, while the actual total visit time was 924 days. Similarly, by modeling the impact of press releases, we predicted 150.11 days of additional play; the actual figure was 142.14 days.

To apply the model and ultimately estimate the work from potential studies, it is important to estimate the size of the initial player cohort and the returning/new player rate. One approach would be to use the actual number of players and returning/new player rate, as measured during the early stages of the project, as estimates. Our experience with Fraxinus has been one of constantly decreasing player numbers, in spite of repeated press releases (*Figure 3C*), and the parameters of our player demographic distribution and the interest in our game have not been such that we received increases in player numbers after the initial release. Fraxinus was a very front-loaded game that did not hold the interest or grow beyond the initial crowd we reached via the media.

Similar decaying trends were also observed with other citizen science projects hosted on Zooniverse (*Ponciano et al., 2014*; *Sauermann and Franzoni, 2015*) and for web searches for trending terms (such as 'Ebola'; *Figure 3—figure supplement 3*), which suggests that the decay in interest observed in crowdsourcing games is similar to that in other topics like the news. This situation is not inevitable; it is possible for the player cohort to grow over time, if the slope of the power law expression in positive player numbers can increase. In practice, this is equivalent to the number of active or new players exceeding those leaving. This could be achieved by an enthusiastic 'viral' growth spread, and the results from the model in this case are similar to those from SIR (susceptible, infected, and recovered) or rumor-spreading models (*Zhao et al., 2013*). However, a viral strategy does not result in unlimited growth of player numbers, when the potential audience is saturated; instead, the amount of work performed begins to decline (*Figure 3C*). The work obtained by a viral strategy does not exceed that by a front-loaded strategy unless somewhat unrealistic growth rate is assumed (e.g., each player invites ten more players on the first day, and there is no restriction on the total available players; *Figure 3C*).

A smaller amount of work than that seen in the front-loaded strategy is done if the players are allowed to defer inviting friends over a time period. We allowed a random time delay of up to 3 months between joining and inviting new players and saw that the overall work was less than if invitation took place in the first 3 days. It is clear that by having a large initial cohort and not adopting a viral strategy, we maximized the work expected from Fraxinus. By applying our model to a proposed crowdsourcing project, it is possible to balance resources (such as public outreach, time taken to analyze data, and initial cohorts of contributors) in order to make the most of potential contributors and to determine whether the approach is likely to be feasible and worthwhile.

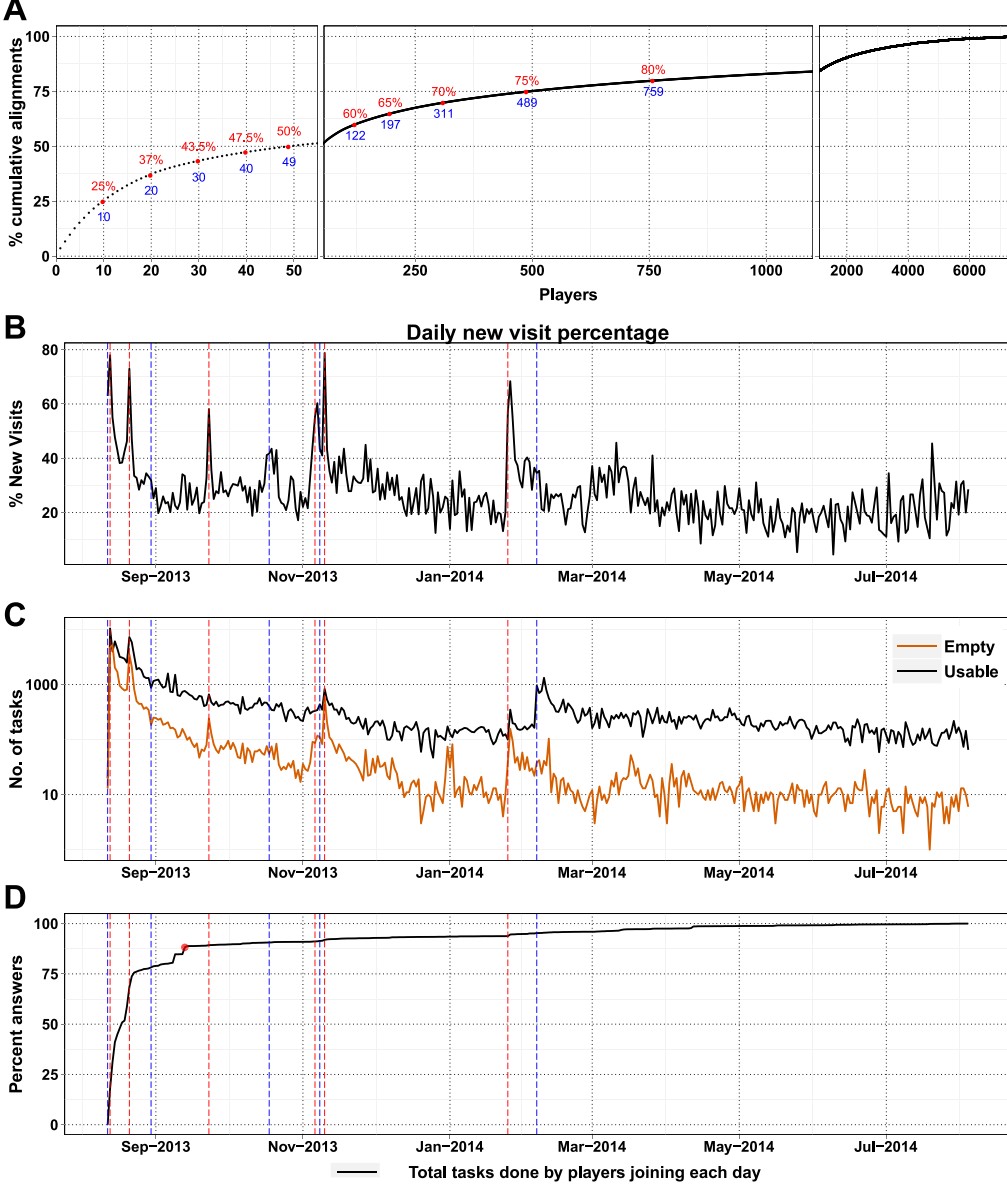

**Figure 2**. High-scoring players and press releases. (**A**) Cumulative contribution by players is plotted against player rank (based on the number of useful answers the player contributed): the ten best players contributed 25% of useful answers. (**B**) Percent of new visits received daily to Fraxinus vs date, with dashed red lines representing press releases and dashed blue lines representing mention on social media. (**C**) Number of usable (black line) and empty (orange line) tasks provided by players vs date. Press releases led to prominent peaks in the number of empty tasks and less prominent peaks in the number of usable tasks. (**D**) Cumulative contribution (by all players) vs date: 88% of the answers were provided within the first month (red dot).

The following figure supplements are available for figure 2:

**Figure supplement 1**. NPs and returning players (RPs) in Fraxinus.

**Figure supplement 2**. Characteristics of players joining on press release dates.

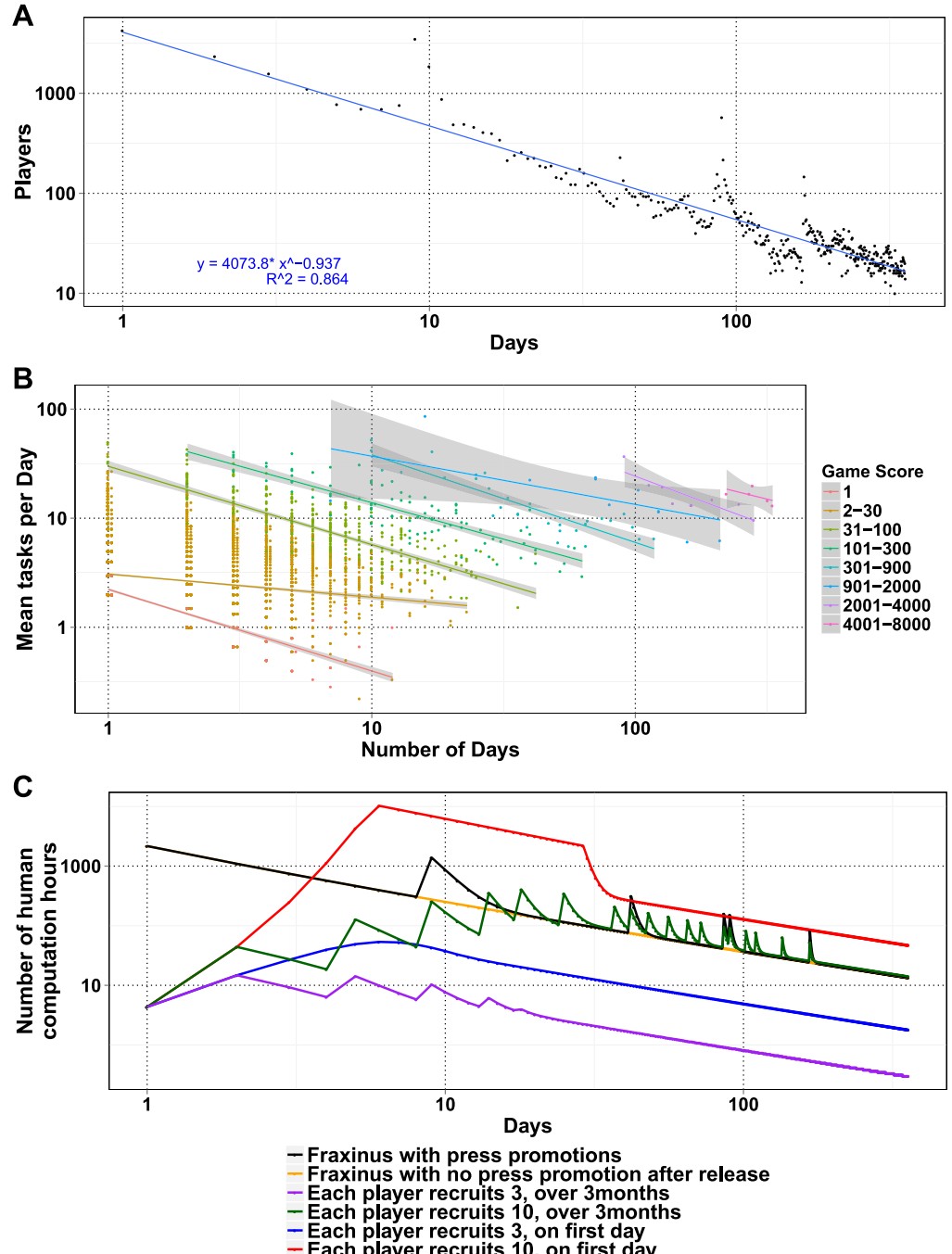

**Figure 3**. Modeling human computation for citizen science projects. (**A**) Actual data showing number of players on each day (y-axis) vs time in days since game release (x-axis) for Fraxinus. The observed distribution of players visiting the game page daily is fitted to a linear model on log scales. (**B**) The enthusiasm of players decreases irrespective of their success at the game. The mean number of tasks completed per day (y-axis) is plotted against the number of days, the players were active (x-axis) for groups of players in similar score groups (color coded based on their scores); contribution decreases over time for all groups. (**C**) Predictions from a model that predicts work done (measured in computation hours; y-axis) as a function of days since game release (x-axis) for six different scenarios: Fraxinus with press releases at and after launch (black); with a press release at launch, but no subsequent press releases (orange); each player recruits three new players (NPs) over a period of 3 months (purple); each player recruits 10 NPs over a period of 3 months (green); each player recruits three NPs on first day (blue); and each player recruits 10 NPs on first day (red).

*Figure 3. continued on next page*

*Figure 3. Continued*

The following figure supplements are available for figure 3:

**Figure supplement 1**. Daily distribution of Fraxinus game visits.

**Figure supplement 2**. Task time line of top 120 players with more than 100 useful answers.

**Figure supplement 3**. Change in interest in the search term 'Ebola' from Google (http://www.google.co.uk/trends/explore#q=ebola&date=4%2F2014%2010m&cmpt=q&tz=) and the predictions for the same from the linear model derived.

## Conclusions

Fraxinus posed a problem that non-specialists were able to positively contribute to. From the patterns of access and return, we observed that the amount of work done is limited to a surprisingly small fraction of contributors, particularly in view to the number of people who have volunteered effort. In the case of Fraxinus, we have been able to build up a small community of skilled users who are willing to collaborate and contribute to our goals. However, it is clear that scientists wishing to take advantage of crowd-sourcing for citizen-science projects must be extremely focused to get value out. Our model of the human computing power that is available for a citizen-science project provides a guide for the design and implementation of future projects.

## Materials and methods

### Fungal materials and sequences

Kenninghall Wood1 (KW1) is an isolate of *H. fraxineus* that was collected and isolated from Kenninghall Wood, Norfolk, UK. KW1 DNA was isolated, and 251 bp paired-end genomic library sequenced using Illumina Miseq. KW1 draft version1 (v1) of genome was assembled using ABySS 1.3.4. Further details of KW1 library preparation and genome assembly can be found from *Saunders et al. (2014)*. Three samples of infected ash branches collected from Ashwellthorpe wood (AT) and Upton broad and Marshes (UB) are referred to as AT1, AT2, and UB1, respectively. RNA was isolated from pith material of infected branches using an RNeasy Plant Mini kit (Qiagen, Manchester, UK). RNA-seq library was prepared using Illumina Truseq kit with 200 bp insert size. Paired-end RNA-seq was carried out on Illumina GAII with read length of 76 bp.

### Variant detection

Paired-end RNA-seq libraries from three samples AT1, AT2, and UB1 were used for variant detection against the KW1 version1 genome assembly. Paired-end RNA-seq reads were aligned to KW1 v1 contigs using BWA (*Li and Durbin, 2009*) mem (v0.7.4) with default settings. SAMTools (*Li et al., 2009*) (v0.1.17) was used to generate sorted BAM files; mpileup and bcftools (view -vcN) commands were used to generate variants. Sequence reads with mapping quality scores less than 20 were ignored in variant selection. Variants called at positions where reference base was unknown were excluded. Positions selected to load on Fraxinus version 1 must have had a minimum coverage of 10. All RNA-seq sequences, BAM, and variant VCF files generated in this study were submitted to the European Nucleotide Archive (ENA) (http://www.ebi.ac.uk/ena) under accession number PRJEB7998.

### Fraxinus game setup

The Fraxinus game interface presents a 21 base reference at the top, and nucleotides are displayed in leaf shape with following colors—green (A), red (T), yellow (G), orange (C), and gray (N). Puzzles were populated with 2–20 reads to be aligned by players. The following scoring scheme was employed with integer values: match score = 5, mismatch score = −3, gap opening score = −5, and gap extension score = −2. Upon joining each new player was taken to an introduction on ash dieback and its impact and subsequently to a tutorial that explains how game play can progress. Players completing the tutorial were awarded one point. Players were then presented with a game index page listing various options, such as choosing a puzzle to solve, the leader board top players, leader board of friends, and any notification.

For each puzzle, players needed to realign the reads against a reference sequence, score as high as they could, and then submit their answer. Players received a point for each answer they submitted and an additional point for submitting the answer with the highest score. If another player beat their score and submitted an answer with a new highest score, they would then get the additional point transferred from the previous player along with a point for submission of answer—this event is referred as 'stealing'. However, if a player matched a previous high score, they did not receive the bonus point. Therefore, we credited only the first player, who provided the highest scoring alignment. Players did not get an extra point for improving their own score on a game, until someone else had 'stolen' the puzzle (by getting a higher score). Each answer submitted by players was stored in a database for subsequent analysis. Player activities, such as accessing the introduction, completion of the tutorial, puzzles accessed, and answers submitted, were stored in a database.

### Variants included in Fraxinus game

We selected 1000 SNP variants and 160 INDEL variants from the RNA-seq analysis of three infected samples (*Table 1*). Variant position and 10 bases on either side were used to extract sequence read information from BWA alignment BAM file. Extracted reads were used to generate one or more puzzles with a maximum of 20 reads per puzzle; resulting in 9022 puzzles from 1000 SNP variants and 1065 puzzles from 160 INDEL variants. The game database was uploaded with generated puzzles that included reference sequence name, 21 base reference sequence hosting the variant, variant position, and details of reads included to realign. Alignment positions were randomly scrambled for included reads, so that players could realign, independent of the information from software alignments.

### Puzzle alignment comparison

For 10,087 puzzles, we received 154,038 answers, of which 35,921 puzzles were empty. Further details on the number of answers are provided in *Supplementary file 1*, Table S1. As reads were selected with in the 21 base window (10 bases either side of the variant position), there were reads that did not cover the variant allele and led to misalignments by the player within the window (*Figure 1—figure supplement 3*). Therefore, we focused our comparison using only reads covering the variant allele according to software alignments and restricted our analysis to 7620 puzzles carrying these reads. The absolute match/mismatch score ratio used in our game is 1.67 (5/3) expected a conservation of 50% between read and the reference and is higher than 0.33, 0.5, and 1, which are used for 99%, 95%, and 75% conservation between reference and read sequences (*States et al., 1991*). Therefore, to be comparable between the alignments by players and software, we realigned the BWA alignments using the scoring system employed in Fraxinus game (BWA options -A 5 -B 3 -O 5 -E 2). Player alignments were stored in CIGAR format (*Li et al., 2009*) with associated alignment start position. However, player alignment start position was set to start of the read, regardless of whether a read was soft clipped or not. So this has been corrected in the analysis. Each puzzle reference nucleotide position was taken randomly from software read alignment information. Therefore, reference sequence position information was corrected from the selected input variants data. For initial comparison of puzzles, position-corrected player alignment CIGARs were compared with BWA CIGAR strings from score adjusted BWA realignments. Then alignments were categorized as similar or different to software based on all high-scoring player outcomes for each puzzle. For each puzzle, the percent of high-scoring players aligned different software was calculated, in addition to percent of reads contributing to the difference. Database dump, data analysis scripts, and source are made available at the authors github (repository Fraxinus version1 data analysis).

### Read alignment comparison

Puzzles, which had all the high-scoring player alignments different to software, were selected, and individual read alignments from these puzzles were extracted and compared to the software read alignments. Individual read alignments were compared in two ways: (1) by calculating percent identity between read and the reference; (2) by calculating the alignment score employed in the Fraxinus game. Both percentage and score calculations for alignments were computed for the bases with in the 21 base game window and for whole read alignments. Any gaps opened and extended were considered as a mismatch in calculating percent identity between read and reference. Alignment score calculation within the game window was normalized per base to be comparable between software and player alignments.

### Game page visit analysis

We used Google Analytics (GA) to record visits and study daily activity to the game page. The open source Ruby API for GA (Google api ruby client samples) was used to extract analytics information regularly. Extracted data included details, such as daily number of player visits, number of new players and returning players, percent new players and returning players, number of new players and returning players, mean duration of daily visit, and mean duration of visit by each player type. We extracted information about geographical distribution of the visits and number of players from each country. Player activity information from 12 August 2013 to 4 August 2014 (358 days) was used to generate reports about the trend of player visits and duration of visits by player type and player geographical distribution. GA uses persistent cookies to identify each player, which may result in an overestimation of number of unique players, especially if the cookies are deleted from the machine. From the comparison of number players registered in the Fraxinus game database and number of players counted from GA, we found that on average 7.25% of additional number of players were recorded by GA. Based on the number of new players and returning players visiting from the Fraxinus database, we estimated that each returning player visits 2.28 times per day. Based on the mean duration of new and returning player visit length, we estimated that 12.5 min was the duration by new players and 29.65 min was the duration spent by each returning player per visit.

### Impact of press releases

To asses the impact of press releases, players joining on or up to 3 days from a press release date were selected, and the total number of useful answers provided by them until 4 August 2014 was pooled. Similar analysis was done for players joining on remaining dates to compute control day player contributions. To calculate number of new and returning players resulting from press releases, the numbers of new and returning players from the selected dates were subtracted by the mean of respective player numbers from 10 previous days.

### Human contribution estimation

We used the data from Fraxinus to combine the play parameters into an equation that can calculate the productivity of a crowdsourcing project. A number of new players and returning players visiting Fraxinus daily were used to fit a power law relationship of $y = ax^b$; where $y$ = player number, $a$ = cohort of players on day 1, $x$ = time since release, and $b$ = rate of decay of players. We have fit separate equations for the number of new players, $NP = ax^{-1.303}$; and the number of returning players, $RP = (a/5x)^{-0.764}$. A parameter $z$ was included to increase the value of NP at arbitrary points to simulate the effect of media or outreach attention. The actual work done by a crowdsourcing project depends on the time taken to complete each task and can be calculated,

$$W = \frac{H - E}{T},$$

where $W$ = Total human computation contribution, $H$ = human computation time, $E$ = education cost (total number of players * tutorial length), $T$ = task length. And H is calculated using following equation,

$$H = NC_iT_n + \sum_{d=2}^{f} NT_n(C_n + C_id^{\alpha}) + ET_r\left(C_rC_id^{\beta}\right),$$

$C_i$ = initial cohort of players (5000), $C_r$ = fraction of the cohort of players returning (1/5), $C_n$ = new players joining due to press release as a function of time, $d$ = day of game, $f$ = end day (358), $T_n$ = mean processing time contributed by a new player in minutes (12.5 min), $T_r$ = mean processing time contributed by a returning player in minutes (70 min), $\alpha$ = rate of decay of new players (−1.303), $\beta$ = rate of decay of returning players (−0.764), $N$ = effectiveness of time contributed by new players (1), $E$ = effectiveness of time contributed by returning players (1).

We provide code implementing the model as used in these analyses at http://nbviewer.ipython.org/github/shyamrallapalli/fraxinus_version1_data_analysis/blob/master/fraxinus_visits_model/Players-nonplayers.ipynb.

### Acknowledgements

We thank Edward Chalstrey, Martin Page, Joe Win, Chris Bridson, and Chris Wilson for technical assistance, and Jodie Pike for Illumina sequencing assistance. We also thank Zoe Dunford and John Innes Centre External Relations team for media outreach and for promoting the game. We also wish to thank all the Fraxinus players who were too modest to be included as co-authors: all contributions are valuable and all are equally appreciated.

## Funding

| Funder | Grant reference | Author |
|--------|-----------------|--------|
| Biotechnology and Biological Sciences Research Council (BBSRC) | Nornex | Dan MacLean |
| Department for Environment, Food and Rural Affairs (Defra) | Nornex | Diane GO Saunders |
| Gatsby Charitable Foundation | None | Dan MacLean |
| John Innes Foundation | None | Dan MacLean |
| The Genome Analysis Centre | National Capability Grant | Dan MacLean |
| John Innes Foundation | Emeritus Fellowship | J Allan Downie |

The funders had no role in study design, data collection and interpretation, or the decision to submit the work for publication.

## Additional files

### Supplementary file

• Supplementary file 1. Table S1: details of alignment answers received for the Fraxinus version 1. Table S2: comparison of the alignments that were aligned differently to software by players. Table S3: details of news articles, press releases, and social network mentions promoting Fraxinus game. Table S4: details about most active players contributing to Fraxinus. Table S5: list of players agreed to be included as authors under 'Fraxinus Players'.

### Major dataset

The following dataset was generated:

| Author(s) | Year | Dataset title | Dataset ID and/or URL | Database, license, and accessibility information |
|-----------|------|---------------|----------------------|--------------------------------------------------|
| Saunders DGO, Yoshida K, Edwards A, Collin JS, Downie A, Kamoun S, MacLean Dan | 2014 | RNA sequencing and variant discovery in the ash dieback pathogen (Hymenoscyphus fraxineus), using several infected samples collected from woodlands of Norfolk, UK | http://www.ebi.ac.uk/ena/data/view/PRJEB7998 | Publicly available at the EBI European Nucleotide Archive (Accession no: PRJEB7998). |

## Author contributions

GR, Analyzed the data from game; Linear equation modelling; Wrote the paper; Prepared data for submission to databases and did the submission, Acquisition of data, Analysis and interpretation of data, Drafting or revising the article; FP, Performed the alignments through game, Acquisition of data; DGOS, KY, AE, SC, BC, MC, DS, MC, Acquisition of data, Contributed unpublished essential data or reagents; CAL, Analysis and interpretation of data; JAD, SK, Conception and design, Contributed unpublished essential data or reagents; TC, Conceived and designed the Fraxinus game, Conception and design; DML, Conceived and designed the Fraxinus game; Conceived and designed the data collection; Samples collected and prepared data for experiment; Linear equation modelling; Wrote the paper, Conception and design, Acquisition of data, Analysis and interpretation of data, Drafting or revising the article

**Ghanasyam Rallapalli** The Sainsbury Laboratory, Norwich, United Kingdom

**Fraxinus Players** Facebook, Fraxinus - Ash Dieback Game Community, Online, United Kingdom

**Diane GO Saunders** The Sainsbury Laboratory, Norwich, United Kingdom; John Innes Centre, Norwich, United Kingdom and The Genome Analysis Centre, Norwich, United Kingdom

**Kentaro Yoshida** The Sainsbury Laboratory, Norwich, United Kingdom

**Anne Edwards** John Innes Centre, Norwich, United Kingdom

**Carlos A Lugo** The Sainsbury Laboratory, Norwich, United Kingdom

**Steve Collin** Norfolk Wildlife Trust, Norwich, United Kingdom

**Bernardo Clavijo** The Genome Analysis Centre, Norwich, United Kingdom

**Manuel Corpas** The Genome Analysis Centre, Norwich, United Kingdom

**David Swarbreck** The Genome Analysis Centre, Norwich, United Kingdom

**Matthew Clark** The Genome Analysis Centre, Norwich, United Kingdom

**J Allan Downie** John Innes Centre, Norwich, United Kingdom

**Sophien Kamoun** The Sainsbury Laboratory, Norwich, United Kingdom

iD http://orcid.org/0000-0002-0290-0315

**Team Cooper** Coopermatic Ltd, Team Cooper, Sheffield, United Kingdom

**Dan MacLean** The Sainsbury Laboratory, Norwich, United Kingdom

**Competing interests:** The authors declare that no competing interests exist.

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
