## [Decision Letter]

Thank you for sending your work entitled “Fraxinus: a citizen science DNA alignment game where players beat software & reveal a winning strategy for crowdsourcing” for consideration at *eLife*. Your article has been favourably evaluated by three reviewers, but needs to be revised in response to the following comments from the referees before we can reach a final decision on publication.

Summary:

The paper, through the analysis of data arising from the platform game Fraxinus, proposes a linear model that predicts the likely outcomes of crowdsourcing projects. The authors, by means of a descriptive analysis, show a case in which the crowd shows higher performances than the machine. Predicting the outcomes of a crowdsourcing project is certainly an interesting topic whose implications are relevant not only for scientists themselves, but also for R&D managers, administrators of granting agencies, and policy makers. The article describes a novel crowdsourcing project and provides relevant insights for crowdsourcing in bioscience. Nonetheless, there is considerable scope for the manuscript to be improved.

Essential revisions:

1) The title of this paper describes Fraxinus as “a citizen science DNA alignment game where players beat machines” and the Abstract states that Fraxinus players “matched or improved computational alignments in 94% of cases”. However, it turns out that only 15% of the time players gained a higher percent identity (and in 78% of alignments the percent identity was equal). Therefore, the claims in the title and Abstract need to be toned down.

The authors should also slightly shorten the paragraph containing the comparison of performances between the crowd and the machine. This topic, despite its relevance, has been already addressed by literature (e.g. [3]).

2) In the Abstract the authors claim that their model will provide a framework for the design and implementation of future citizen-science initiatives. However, the paper just provides limited hints of such a framework. The authors should consider extending this section. The authors should also consider adding a paragraph about the generalizability of the model, possibly containing evidences of its goodness (e.g. through the prediction of the outcomes of other crowdsourcing project).

3) Many key concepts necessary to describe multiple sequence alignments and genetic variant predictions are explained only brief. Please supply more detailed descriptions for the benefit of readers who are not familiar with these concepts.

4) The authors should explain their model more fully. In particular, they should better explain the model assumptions (e.g. why press releases only influence the behavior of new players?) and the implications of these assumptions. Crowston and Fagnot (2008), Panciera, Halfaker and Terveen (2009), and Jackson and colleagues (2015WP) develop dynamic models of virtual collaboration that may be useful in thinking about this issue.

5) The authors should consider a refinement of their model to address the following points:

Results and Discussion, first paragraph: The authors claim that 49 players (0.7%) contributed to half of the total answers. However, the authors do not consider this information when they developed their model.

Similarly, they write (Results and Discussion, fifth paragraph) that not all time contributed by players is equally productive. However, they make this assumption in their model. The authors should explain the reasons of these choices.

A number of recent studies (e.g. [14]) describes user's contribution pattern and highlights that the idea of a homogenous group of participants is far from reality. The authors should consider these contributions to refine their model.

Using Google Analytics persistent cookies may generate a bias in assessing both the number of new players (overestimation) and the number of returning players (underestimation). The authors should describe how they took into account this bias in their computation.

6) Concerning the results, it would be good to have a justification for the parameters used. Why do you choose a 21 nt game window? Do you try the game with a longer window? Could you expect better results if a longer window is used? In the same sense, the alignments are compared in two ways; by calculating the percent identity and by computing an alignment score. With respect to the measure of identity, why the authors did not use ‘standard’ measures such as Sum of Pairs, and Total Column metrics? What about comparing the score of the alignments using the score functions reported by state of the art MSA algorithms?

7) It is not clear to the reader what software/computational method the players are being compared to, especially in the figures and tables. Please explicitly the software/computational method in the captions of the relevant figures and tables.

8) The section ‘Human contribution estimation’ needs some revision. If the formulae given in this section are standard formulae taken from the literature, please supply references. And if these formulae are not standard formulae, please supply more information about how they were derived.

9) Please consider adding the following references and discussing them in the text at the appropriate place:

Open-Phylo: a customizable crowd-computing platform for multiple sequence alignment Genome Biology 14, R116 (2013). This paper seems to have a lot of common features with the study presented here, and it can be used for comparison purposes.

Algorithm discovery by protein folding game players PNAS 108, 18949-18953 (2011). This paper could be used to develop the idea that the alignment strategies used by players could be implemented in algorithms for potential improvements.

It would be useful to be explicit about the relationship between the present work and previous publications (MacLean et al., GigaScience, 2013; MacLean et al., *eLife*, 2013).

---

## [Author Response]

*1) The title of this paper describes Fraxinus as “a citizen science DNA alignment game where players beat machines” and the Abstract states that Fraxinus players “matched or improved computational alignments in 94% of cases”. However, it turns out that only 15% of the time players gained a higher percent identity (and in 78% of alignments the percent identity was equal). Therefore, the claims in the title and Abstract need to be toned down*.

We think the claim is justified in the Abstract. As the reviewers point out, in 78% of cases the algorithms and players are equally good, in 15% of the remaining the players are better – hence “matched or improved” – and we think this large proportion of equaling and bettering is a substantial. We would be happy to rephrase this in the title: “Fraxinus: a citizen science game where players can improve software DNA alignments & a model for strategising in crowdsourcing”.

*The authors should also slightly shorten the paragraph containing the comparison of performances between the crowd and the machine. This topic, despite its relevance, has been already addressed by literature (e.g.*
[3]*)*.

We take this paragraph to be the second in Results and Discussion that describes the specific findings on performance between player and machine in Fraxinus. We have edited to reduce the word count and removed the mention of gene functions. Cooper et al. describe FoldIt has had success with players doing protein structures – not sequence alignments – and do not describe the specific results for DNA sequence alignment as in Fraxinus. It is not a given that the crowd would beat the computer in this different problem domain and it is important that the extent to which crowd and machine differ in Fraxinus specifically is reported. With this paragraph we do simply this and show that the game is a useful tool that fulfilled its primary purpose. There isn't much more in this paragraph other than some terse statements about player machine differences and we can't edit it much further and retain the important, specific information.

*2) In the Abstract the authors claim that their model will provide a framework for the design and implementation of future citizen-science initiatives. However, the paper just provides limited hints of such a framework. The authors should consider extending this section. The authors should also consider adding a paragraph about the generalizability of the model, possibly containing evidences of its goodness (e.g. through the prediction of the outcomes of other crowdsourcing project)*.

In the Abstract we have replaced the word ‘framework’ with ‘guide’. We have added text describing how the model may be generalized to the discussion in Conclusions and a brief mention in the Abstract.

We agree that showing the goodness of the model through predictions would be useful, however despite making numerous data requests from other published projects we were not able to get user/player visit data from any other crowdsourcing project. We would need these data to verify and calibrate our model and predictions. Hence we can only make predictions based on assumed parameters but not falsify them, which isn't very useful and doesn't help in the aim of showing goodness.

In supplemental material we have added data that do show the model accurately represents the interest and attention on topics as revealed by Google Trends data. We show the decay of interest in the topic of Ebola over recent time and the model's prediction of the same. Games and topics are analogous in the initial interest component of the crowdsourcing so can be taken as a somewhat close proxy of interest in a crowdsourcing game. We have added a new supplementary figure (Figure 3—figure supplement 3) of these data.

*3) Many key concepts necessary to describe multiple sequence alignments and genetic variant predictions are explained only brief. Please supply more detailed descriptions for the benefit of readers who are not familiar with these concepts*.

We have added a paragraph on these concepts to the Introduction.

*4) The authors should explain their model more fully. In particular, they should better explain the model assumptions (e.g. why press releases only influence the behavior of new players?) and the implications of these assumptions*.

Press releases do not only influence the behaviour of new players, we observed that press releases bring new players but data show the returning players increased only minimally, this difference is covered by the model also and described in Figure 2—figure supplement 2 on the behaviour of players that are new versus those returning. It isn't really an assumption since we observed the difference and model it directly.

*Crowston and Fagnot (2008), Panciera, Halfaker and Terveen (2009), and Jackson and colleagues (2015WP) develop dynamic models of virtual collaboration that may be useful in thinking about this issue*.

The studies suggested by the reviewers address virtual collaborations observed during crowdsourcing projects especially Wikipedia. These require expertise in the field, also each article is edited by multiple individuals who virtually collaborate to complete the task. Collaboration between players is not possible in Fraxinus, rather players compete to gain higher scores and they are not required to have knowledge of DNA alignments. The fundamental observations in these references are similar to ours, i.e. that a core set of individuals/players contribute most and these individuals provide regular time contributions. These assumptions are already built in to our linear model.

5) The authors should consider a refinement of their model to address the following points:

Results and Discussion, first paragraph: The authors claim that 49 players (0.7%) contributed to half of the total answers. However, the authors do not consider this information when they developed their model.

These are already incorporated, we model different decay rate for returning and new players and the model doesn't explicitly create player agents that return work but rather models the change in number of players. Hence the model is a model of time donated and therefore by proxy the work done. It is therefore not necessary be so explicit and our model can remain generic. The amount of work done is assumed to be a function of the time donated.

*Similarly, they write (Results and Discussion, fifth paragraph) that not all time contributed by players is equally productive. However, they make this assumption in their model. The authors should explain the reasons of these choices*.

Again we do consider this but it isn't a feature of Fraxinus, we didn't find that new players were much worse at the task than returning players. As described the model is one of time donated rather than player agents doing work, and the function describing work done should be adjusted accordingly by the final user. We do not assume constant productivity in our work done calculations, only time spent. The difference in the model is merely a difference between a simple effectiveness factor for each of the new or returning players contributions. In Fraxinus these factors are equal so could be ignored and were not therefore in our description of the Fraxinus parameterised model. We have made this more explicit in the revised description of the model and mention how these can be applied.

*A number of recent studies (e.g.*
[14]*) describes user's contribution pattern and highlights that the idea of a homogenous group of participants is far from reality. The authors should consider these contributions to refine their model*.

We have considered the effects of non-homogenous groups, we have dedicated returning players that donate time and leave at one rate and casually non-returning players that leave at another rate, and we have incorporated the different effectiveness of new and experienced players in the revised description of the model.

*Using Google Analytics persistent cookies may generate a bias in assessing both the number of new players (overestimation) and the number of returning players (underestimation). The authors should describe how they took into account this bias in their computation*.

Since we store each player’s Facebook ID in the database we do not need to rely solely on Google Analytics to identify players. We did compare these, Google Analytics does 7% overestimation and this is explained in the Methods section. As described, the majority of the calculations for estimation of new and returning players have been taken from player activity from our database, not directly from Google Analytics.

6) Concerning the results, it would be good to have a justification for the parameters used. Why do you choose a 21 nt game window?

Although many scientists have big screens to do their work with, in the general public there is a screen size limitation. Most people at time of game design were using computers with around 1440 pixel width. Indeed Google Analytics of player screen resolutions indicate that ∼72% of game plays were accessed on a screen resolution of 1440pixel or smaller and ∼99% game plays on a 1920pixel or smaller screens. To design a game we need to take into account the amount of screen that needs to be taken up by the Facebook window and its features and adverts which left a smaller still section to work with. At design phase 21 nt was the size that fits well given a big enough for a useable and viewable game. The nucleotides on screen had to be big enough to be manipulated as well as seen so graphics size quickly became a problem.

Do you try the game with a longer window?

Not in production, only during development where longer windows proved to be too small to use well.

Could you expect better results if a longer window is used?

Alignments would take longer to do so people would play fewer alignments, so less replication would be done. Some alignments *may* improve but consensus accuracy from replication *may* decrease. I suppose it depends on what are considered better results in this context. Is longer better, or is better?

In the same sense, the alignments are compared in two ways; by calculating the percent identity and by computing an alignment score. With respect to the measure of identity, why the authors did not use ‘standard’ measures such as Sum of Pairs, and Total Column metrics?

It is not correct to assess these alignments as Multiple Sequence Alignments (of which Sum of Pairs and Total Column are metrics). These are serial, and independent pairwise sequence alignments (which has a bearing on the comments below too), each sequence aligned is a read from a genome sequencer, an independent measurement of the sequence, they are not and should not be taken en masse and assessed as an MSA.

The percent identity that we used is an intuitive and straightforward measure that captures the variants well in pairwise sequence alignments and the score we used is the aligner's own score so set a level playing field for the comparison of player and aligner performance.

What about comparing the score of the alignments using the score functions reported by state of the art MSA algorithms?

This is exactly what we did, we used BWA mem, the most widely used and a very modern algorithm that is about state-of-the-art as it gets for sequence read alignment. This is mentioned on the first line in the Methods section.

*7) It is not clear to the reader what software/computational method the players are being compared to, especially in the figures and tables. Please explicitly the software/computational method in the captions of the relevant figures and tables*.

We mention this on the first line in the Results and Discussion section, we have reiterated it later in the main text at an appropriate place, in the captions and in the Methods section.

*8) The section ‘Human contribution estimation’ needs some revision. If the formulae given in this section are standard formulae taken from the literature, please supply references. And if these formulae are not standard formulae, please supply more information about how they were derived*.

This is a standard definite integral, but in there is a typo that makes it hard to read (strictly it made a nonsense of the formula – apologies.) Nonetheless supplying information about the derivation of these isn't really helpful, it is a common sort of formula and we do supply descriptions and values for each variable. That said though, after discussions we feel that a different notation would be more readable, intuitive and avoid the need for deriving the formula. Instead we propose to represent the model as a discrete sum over the days of a crowdsourcing project. This is functionally equivalent to the continuous integral since the model works on whole days anyway and is more in line with the simulations we carried out and the provided code. The variables have all been renamed (hopefully these will be easier to remember) in the new annotation but represent the same things plus the explicitly stated effectiveness factor (ignored previously in Fraxinus because it was 1). We have amended the text to include this new notation. We also provide the code for the model as used in this analysis [http://nbviewer.ipython.org/github/shyamrallapalli/fraxinus_version1_data_analysis/blob/master/fraxinus_visits_model/Players-nonplayers.ipynb] (fraxinus_visits_model).

*9) Please consider adding the following references and discussing them in the text at the appropriate place*:

*Open-Phylo: a customizable crowd-computing platform for multiple sequence alignment Genome Biology 14, R116 (2013). This paper seems to have a lot of common features with the study presented here, and it can be used for comparison purposes*.

The Open-Phylo project is an elaboration of the original Phylo project (which we cite). Open-Phylo extends the Phylo project and this paper describes an exciting new crowd computing platform which scientists can use to upload their own puzzles for solution by the crowd. It does not describe the results from the Phylo multiple sequence alignment citizen sequence project (in the original, cited paper). We believe that the paper cited is the best one for comparison.

*Algorithm discovery by protein folding game players PNAS 108, 18949-18953 (2011). This paper could be used to develop the idea that the alignment strategies used by players could be implemented in algorithms for potential improvements*.

We have developed the idea slightly in Results and cited the method described in this paper.

*It would be useful to be explicit about the relationship between the present work and previous publications (MacLean et al., GigaScience, 2013; MacLean et al.,* eLife*, 2013)*.

We have clarified briefly the content of each of these when cited.